# Use and experiences of galactagogues while breastfeeding among Australian women

Grace M. McBride[1,2,3], Robyn Stevenson[1], Gabriella Zizzo[1], Alice R. Rumbold[1,2,3], Lisa H. Amir[4,5], Amy K. Keir[1,2,3], Luke E. Grzeskowiak[1,2,3,6,7] *

**1** Adelaide Medical School, University of Adelaide, Adelaide, Australia, **2** The Robinson Research Institute, University of Adelaide, Adelaide, Australia, **3** South Australian Health and Medical Research Institute, Adelaide, Australia, **4** Judith Lumley Centre, La Trobe University, Melbourne, Australia, **5** Breastfeeding Service, Royal Women's Hospital, Parkville, Australia, **6** SA Pharmacy, SA Health, Adelaide, Australia, **7** Flinders Health and Medical Research Institute, Flinders University, Adelaide, Australia

* luke.grzeskowiak@flinders.edu.au

## Abstract

### Background

Galactagogues are substances thought to increase breast milk production, however evidence to support their efficacy and safety remain limited. We undertook a survey among Australian women to examine patterns of use of galactagogues and perceptions regarding their safety and effectiveness.

### Methods

An online, cross-sectional survey was distributed between September and December 2019 via national breastfeeding and preterm birth support organisations, and networks of several research institutions in Australia. Women were eligible to participate if they lived in Australia and were currently/previously breastfeeding. The survey included questions about galactagogue use (including duration and timing), side effects and perceived effectiveness (on a scale of 1 [Not at all effective] to 5 [Extremely effective]).

### Results

Among 1876 respondents, 1120 (60%) reported using one or more galactagogues. Women were 31.5 ± 4.8 years (mean ± standard deviation) at their most recent birth. Sixty-five percent of women were currently breastfeeding at the time of the survey. The most commonly reported galactagogues included lactation cookies (47%), brewer's yeast (32%), fenugreek (22%) and domperidone (19%). The mean duration of use for each galactagogue ranged from 2 to 20 weeks. Approximately 1 in 6 women reported commencing galactagogues within the first week postpartum. Most women reported receiving recommendations to use herbal/dietary galactagogues from the internet (38%) or friends (25%), whereas pharmaceutical galactagogues were most commonly prescribed by General Practitioners (72%). The perceived effectiveness varied greatly across galactagogues. Perceived effectiveness was highest for domperidone (mean rating of 3.3 compared with 2.0 to 3.0 among other

**Data Availability Statement:** Data cannot be shared publicly because the ethics committee restricts secondary use of the data currently. Data are available from The University of Adelaide

Human Research Ethics Committee (contact T: +61 8 8313 5137 | F: +61 8 8313 3700 | research. services@adelaide.edu.au) for researchers who meet the criteria for access to confidential data.

**Funding:** GM was supported by an Australian Government Research Training Program Scholarship. AK was supported by a National Health and Medical Research Council Early Career Fellowship (GNT1161379). LG receives salary support through a Mid-Career Research Fellowship provided by The Hospital Research Foundation (C-MCF-10-2019). LA, AR, GZ and LG were awarded a Robinson Research Institute Engaging Opportunities Grant 2019. The funders had no role in study design, data collection and analysis, decision to publish, or preparation of the manuscript.

**Competing interests:** The authors have declared that no competing interests exist.

galactagogues). Over 23% of domperidone users reported experiencing multiple side effects, compared to an average of 3% of women taking herbal galactagogues.

## Conclusions

This survey demonstrates that galactagogues use is common in Australia. Further research is needed to generate robust evidence about galactagogues' efficacy and safety to support evidence-based strategies and improve breastfeeding outcomes.

## Introduction

Breastfeeding is widely recognised to promote lifelong health for both the mother and infant [1]. International recommendations are exclusive breastfeeding until six months of age, with ongoing breastfeeding for two years or longer [2, 3]. In Australia, evidence indicates that the majority of women initiate breastfeeding at birth; however, by six months of age, only 60% are providing any breast milk, and 16% are exclusively breastfeeding [4]. This marked drop in exclusive breastfeeding has been observed in many other high-income countries [5]. Previous research shows that lactation insufficiency (also referred to as low breast milk supply), whether real or perceived, is one of the most common reasons women discontinue breastfeeding [6, 7]. Lactation insufficiency can be caused by several factors, including insufficient mammary tissue, irregular hormone levels, and ineffective milk removal from the breast [8].

The first-line management of lactation insufficiency involves non-pharmacological interventions, such as ensuring correct infant positioning and attachment [8, 9]. Where lactation insufficiency persists, galactagogues—the term used to describe substances thought to promote or increase breast milk production—may be used. Commonly reported galactagogues include dietary or herbal supplements, for example, oats or fenugreek, and pharmaceutical treatments such as domperidone [10]. Anecdotally, recent studies demonstrate widespread awareness and use of there is increased promotion of dietary galactagogues such as lactation cookies [11, 12]. An examination of widely promoted recipes and commercially available products indicates that lactation cookies contain highly variable combinations and quantities Internet searches outline a variety of ingredients, including oats, brewer's yeast and flaxseed.

A recent Cochrane review on the use of oral galactagogues for increasing breast milk production in mothers of non-hospitalised term infants identified forty-one randomised clinical trials [10]. The review found uncertain evidence that galactagogues improve breast milk volume or longer-term breastfeeding outcomes [10]. In contrast, several high-quality studies have found domperidone effective in increasing breast milk production, specifically among mothers of preterm infants [13, 14]. However, the use of domperidone remains controversial. Domperidone use at doses above 30 mg daily may present a risk of serious cardiac side effects [15]. However the relevance to breastfeeding women has been questioned as previous data on increased cardiac risks mainly involved males and those aged over 60 years [16].

The considerable variation across studies concerning study population, intervention type, and outcome evaluation has led to ongoing treatment uncertainties. This is reflected in the recent guidelines issued by the Academy of Breastfeeding Medicine, which state that there is insufficient evidence to recommend one galactagogue over another [17].

Despite conflicting evidence regarding the benefits of galactagogues in clinical practice, there is evidence that breastfeeding women commonly use galactagogues, and use may be increasing. For example, a 2012 Australian survey of 304 breastfeeding women observed that

24% of respondents reported using a herbal galactagogue [18]. Estimates of uptake of the pharmaceutical galactagogue domperidone appear more variable. Studies based on prescribing/dispensing records from Australia, Canada and the UK show increasing trends in use, with varying overall prevalence of use ranging from 2.7% to 20% [19–22]. In specific populations such as following preterm birth, prevalence appears even higher, up to 30% [19, 21]. Further, Grzeskowiak *et al.* examined queries relating to galactagogues at an Australian medicines information centre from 2001 to 2014 that demonstrated a significant trend towards increased phone calls regarding herbal galactagogues (0% to 23% of calls regarding galactagogues from 2001 to 2014) compared with a consistent interest in pharmaceutical galactagogues [23]. Unfortunately, the most recent studies evaluating galactagogue use only include data until 2015 and did not collect data on all types of galactagogues [12, 18–20]. Therefore, we sought to undertake a survey to examine patterns of use of galactagogues, women's experiences relating to use, as well as their perceptions regarding effectiveness.

## Methods

### Ethics

This study was approved by the Human Research Ethics Committee at the University of Adelaide (approval number H-2019033934).

### Survey administration

Women currently living in Australia and either currently breastfeeding or who had previously breastfed were eligible to complete the survey. The survey was available online between 27 September 2019 and 12 December 2019. The survey consisted of part A, perceived safety and knowledge of galactagogues, and part B, personal experiences and use of galactagogues, including the self-perceived effectiveness, side effects and duration of use. If women had not taken any galactagogues, they did not complete part B of the survey. This paper will focus predominantly on part B of the survey. Questions included in the survey covered the timing and duration of use of substances, sources of recommendation, side effects experienced and perceived effectiveness. The perceived effectiveness of galactagogues was assessed using a 5-point Likert scale from 1 (Not at all effective) to 5 (Extremely effective). The survey was tested for face validity with two consumers and an academic breastfeeding expert. Only minor changes were made to the survey before formal distribution through social networks (i.e. Facebook, Twitter, email) of the Australian Breastfeeding Association [24, 25] (Australia's national breastfeeding support service, assisting more than 80,000 women each year, with over 1100 breastfeeding counsellors available), Miracle Babies [26] (Australia's leading organisation supporting premature and sick newborns, present in 143 Neonatal Intensive Care Units or Special Care Nurseries in Australia), as well as research networks of the author's respective institutions (e.g. The Robinson Research Institute, and The University of Adelaide). Participants were encouraged to share the survey and post links to the survey through their own social networks. The survey was piloted with a small group of consumers (reviewed by representatives from the Australian Breastfeeding Association and Miracle Babies) and academic experts in survey design, resulting in minor modifications before the final survey was launched. The complete survey is available as S1 File.

Study data were collected and managed using Research Electronic Data Capture (REDCap) hosted at The University of Adelaide [27, 28]. REDCap is a secure, web-based software platform designed to support data capture for research studies, providing 1) an intuitive interface for validated data capture; 2) audit trails for tracking data manipulation and export procedures; 3) automated export procedures for seamless data downloads to standard statistical

packages, and 4) procedures for data integration and interoperability with external sources. Only study investigators involved in the study had access to the data.

Completing the survey was voluntary, and no incentives were offered to participants. Respondents had the opportunity to submit their responses anonymously or could choose to include their contact details. When contact details were provided, respondents were approached to participate in a separate qualitative study investigating women's experiences of using galactagogues. Only those who provided their contact details were able to withdraw their responses, however none elected to withdraw their responses. A total of 2152 responses were received, 7 responses were removed due to suspected duplicate entries based on identical maternal characteristics provided in the entry section, and a further 90 were removed due to births occurring outside of Australia.

### Data analysis

Data were cleaned and analysed using STATA 14 (StataCorp LP, College Station, TX). Graphical images were produced using GraphPad Prism version 9 (GraphPad Software, La Jolla California USA) and R Upset Package [29]. Maternal demographic characteristics and data on use and experiences of galactagogues were described using descriptive statistics. The most common combinations of galactagogues used were graphed using an UpSet plot. Differences in maternal characteristics according to any galactagogue use were compared using Student's T-test for means and Pearson's Chi$^2$ test for categorical variables. Duration of use was reported separately for each galactagogue according to those that were continuing use at the time of the survey and those that had stopped using it prior to completing the survey. Descriptive statistics were used to report the means and standard deviations. Where data were non-normally distributed, the median and inter-quartile ranges were used. Statistical significance was defined as a $P < 0.05$.

## Results

A total of 1876 women responded to the survey. Maternal demographic characteristics of survey respondents are presented in Table 1. Briefly, the average age of women who responded was 31.5 years old, while most had completed secondary schooling or higher (92%) and almost half were primiparous (47%). At the time of the survey, 1217 (65%) of women reported they were currently breastfeeding their infant. For women who reported currently breastfeeding, the average infant age at the time of survey response was 10.7 months (mean ± 10 months standard deviation). Women who reported having ceased breastfeeding before completing the survey discontinued at an average of 21 months (mean ± 11 months standard deviation). Almost half of all respondents (49%) felt they could not produce enough breast milk for their child, and 63% sought help from a lactation consultant or breastfeeding expert. Of women who had stopped breastfeeding prior to completing the survey (35%), 19% reported stopping due to low milk supply.

### Galactagogue use

Overall, 60% of women (n = 1120) reported taking one or more galactagogues during breastfeeding. Women who had preterm births, saw a lactation consultant, were primiparous, had perceived low milk supply, had a Caesarean section, or required supplemental feeding with infant formula were more likely to use galactagogues (Table 1).

Information on individual galactagogue use is presented in Table 2. The most commonly used galactagogue included lactation cookies (47%), brewer's yeast (32%) and fenugreek (22%). The use of 'Other' galactagogues were reported by 7.3% (n = 137) of women, which

**Table 1. Maternal characteristics according to any reported use of a galactagogue during breastfeeding.**

|  | Total survey population | Did not use galactagogue | Used a galactagogue | P-value* |
|---|---|---|---|---|
|  | n (%) | n (%) | n (%) |  |
| **N (Total = 1876)** | 2055 | 756 | 1120 |  |
| **Mothers age at delivery (years; mean ± SD)** | 31.5 ± 4.8 | 32 ± 5.2 | 31.2 ± 4.5 | <0.001 |
| **Youngest child's age at survey** |  |  |  | 0.005 |
| 0–< 6 months | 560 (30) | 223 (30) | 335 (30) |  |
| ≥ 6–< 12 months | 370 (20) | 124 (17) | 246 (22) |  |
| ≥ 12 months | 936 (50) | 405 (54) | 527 (48) |  |
| **State/Territory of youngest child's birth** |  |  |  | 0.291 |
| Australian Capital Territory | 88 (5) | 40 (5) | 48 (4) |  |
| New South Wales | 453 (24) | 192 (26) | 259 (23) |  |
| Northern Territory | 23 (1) | 9 (1) | 14 (1) |  |
| Queensland | 322 (17) | 111 (15) | 210 (19) |  |
| South Australia | 378 (20) | 150 (20) | 225 (20) |  |
| Tasmania | 43 (2) | 17 (2) | 25 (2) |  |
| Victoria | 407 (22) | 176 (24) | 231 (21) |  |
| Western Australia | 150 (8) | 55 (7) | 94 (9) |  |
| **Completed secondary school** | 1887 (92) | 698 (93) | 1027 (92) | 0.834 |
| **Primiparous** | 882 (47) | 255 (34) | 625 (56) | < 0.001 |
| **Multiple birth** | 39 (2) | 14 (2) | 25 (2) | 0.578 |
| **Preterm birth** | 218 (12) | 66 (9) | 150 (14) | 0.002 |
| **Caesarean-section** | 621 (33) | 192 (26) | 426 (38) | <0.001 |
| **Perceived low milk supply** | 928 (49) | 162 (22) | 761 (68) | <0.001 |
| **Saw a lactation consultant** | 1184 (63) | 381 (51) | 798 (71) | <0.001 |
| **Supplemented with infant formula** | 561 (30) | 111 (15) | 446 (40) | <0.001 |
| **Any smoking during breastfeeding** | 66 (4) | 23 (3) | 43 (4) | 0.358 |

* Chi$^2$ test between those that used and did not use a galactagogue.

included oats (n = 87; 4.7%), malt products (n = 42; 2.2%), and flaxseed or linseed (n = 13; 0.7%).

With respect to domperidone and metoclopramide, which are only available by prescription, these were most commonly prescribed by general practitioners (76% and 67% respectively), followed by obstetricians/gynaecologists (20% and 10% respectively). For the remaining galactagogues, the most common recommendation source was the internet (ranging from 28–50%) and friends (ranging from 15–45%). Healthcare professionals such as community pharmacists (2–6%), general practitioners (2–7%), and obstetricians/gynaecologists (1–2%) were uncommon sources of recommendation. One in three women taking herbal or dietary galactagogues reported using two or more recommendation sources.

Among those reporting galactagogue use, 27% took only one substance, while 46% used three or more galactagogues. The maximum number of galactagogues used was 10. The most common patterns of galactagogue use are represented in Fig 1. Lactation cookies featured in the top five different combinations of galactagogues used, and were the most used sole galactagogue.

## Timing of commencement of galactagogues

Reported timing of commencement of galactagogue use is presented in Fig 2. Approximately 50% of galactagogues were commenced within the first four weeks postpartum, with 18.5%

**Table 2. Reported use of galactagogues and information sources from breastfeeding women (n = 1876).**

| | Domperidone | Metoclopramide | Fenugreek | Blessed Thistle | Fennel | Milk Thistle | Ginger | Brewer's yeast | Lactation cookies | Combination of herbs |
|---|---|---|---|---|---|---|---|---|---|---|
| **Took substance (n (%))** | 355 (19) | 21 (1) | 421 (22) | 98 (5) | 157 (8) | 40 (2) | 52 (3) | 592 (32) | 884 (47) | 109 (6) |
| **Mothers age at birth (years; mean ± SD)** | 31.8 ± 4.6 | 34.1 ± 4.2 | 31.7 ± 4.4 | 31.7 ± 4.3 | 31.5 ± 4.5 | 31.5 ± 4.5 | 31.2 ± 4.9 | 30.9 ±4.4 | 31 ± 4.4 | 32.1 ± 4.4 |
| **Child's age at survey*** | | | | | | | | | | |
| 0–6 months | 110 (31) | 2 (10) | 105 (25) | 29 (30) | 44 (28) | 13 (33) | 19 (37) | 158 (27) | 264 (30) | 42 (39) |
| 6–12 months | 79 (22) | 4 (19) | 86 (21) | 18 (18) | 43 (28) | 8 (20) | 16 (31) | 135 (23) | 210 (24) | 23 (21) |
| 12+ months | 163 (46) | 15 (71) | 227 (54) | 51 (52) | 68 (44) | 19 (48) | 16 (31) | 292 (50) | 401 (46) | 43 (40) |
| **Maternal characteristics *** | | | | | | | | | | |
| Primiparous | 208 (59) | 7 (33) | 238 (57) | 49 (50) | 91 (58) | 19 (48) | 28 (54) | 332 (56) | 528 (60) | 60 (55) |
| Preterm birth | 74 (21) | 7 (33) | 70 (17) | 17 (17) | 22 (14) | 9 (23) | 7 (13) | 85 (14) | 125 (14) | 14 (13) |
| Caesarean section | 162 (46) | 14 (67) | 157 (37) | 33 (34) | 61 (39) | 18 (45) | 25 (48) | 207 (35) | 328 (37) | 46 (42) |
| Perceived low milk supply | 327 (92) | 19 (90) | 331 (79) | 88 (90) | 114 (73) | 34 (85) | 40 (77) | 423 (71) | 619 (70) | 72 (66) |
| **Took only this substance *** | 41 (12) | 0 | 32 (8) | 0 | 6 (4) | 0 | 3 (6) | 23 (4) | 177 (20) | 11 (10) |
| **Two or more recommendation sources *** | 57 (16) | 1 (5) | 168 (40) | 34 (35) | 43 (27) | 15 (38) | 14 (27) | 272 (46) | 421 (48) | 27 (25) |
| **Who prescribed/recommended *** | | | | | | | | | | |
| General Practitioner | 271 (76) | 14 (67) | 30 (7) | 5 (5) | 3 (2) | 1 (3) | 3 (6) | 22 (4) | 21 (2) | 2 (2) |
| Obstetrician/Gynaecologist | 72 (20) | 2 (10) | 7 (2) | 1 (1) | 1 (1) | | 1 (2) | 4 (1) | 8 (1) | |
| Midwife | 41 (12) | 2 (10) | 67 (16) | 14 (14) | 8 (5) | 3 (8) | 3 (6) | 65 (11) | 87 (10) | 6 (6) |
| Neonatologist/paediatrician | 19 (5) | 2 (10) | 9 (2) | 2 (2) | | | | 1 (0) | 4 (0) | 1 (1) |
| Internet search | | | 119 (28) | 30 (31) | 54 (34) | 20 (50) | 17 (33) | 278 (47) | 356 (40) | 34 (31) |
| Lactation consultant | | | 90 (21) | 26 (27) | 18 (11) | 6 (15) | 3 (6) | 76 (13) | 117 (13) | 9 (8) |
| Friends | | | 87 (21) | 15 (15) | 28 (18) | 11 (28) | 12 (23) | 211 (36) | 395 (45) | 26 (24) |
| Family | | | 58 (14) | 9 (9) | 25 (16) | 5 (13) | 16 (31) | 98 (17) | 174 (20) | 14 (13) |
| Child & family health nurse | | | 45 (11) | 8 (8) | 5 (3) | 2 (5) | 1 (2) | 35 (6) | 60 (7) | |
| Naturopath | | | 40 (10) | 14 (14) | 29 (18) | 8 (20) | 7 (13) | 23 (4) | 17 (2) | 16 (15) |
| Neonatal nurse | | | 36 (9) | 5 (5) | 7 (4) | 4 (10) | 3 (6) | 30 (5) | 54 (6) | 1 (1) |
| Mother's group | | | 33 (8) | 4 (4) | 11 (7) | 2 (5) | 4 (8) | 56 (9) | 112 (13) | 4 (4) |
| Social media | | | 34 (8) | 5 (5) | 11 (7) | 3 (8) | 5 (10) | 91 (15) | 160 (18) | 11 (10) |
| Blogs or online discussion forums | | | 24 (6) | 4 (4) | 6 (4) | 1 (3) | 5 (10) | 56 (9) | 77 (9) | 4 (4) |
| Community pharmacist | | | 24 (6) | 4 (4) | 4 (3) | | | 18 (3) | 15 (2) | 5 (5) |
| Breastfeeding helpline | | | 10 (2) | | 2 (1) | | 1 (2) | 12 (2) | 20 (2) | 2 (2) |
| Books | | | 6 (1) | 1 (1) | 3 (2) | | | 8 (1) | 7 (1) | 2 (2) |
| Podcasts | | | 1 (0) | | | | 1 (2) | 1 (0) | 3 (0) | 1 (1) |
| Other | 15 (4) | 2 (10) | 18 (4) | 7 (7) | 14 (9) | 3 (8) | 1 (2) | 21 (4) | 44 (5) | 14 (13) |

* (n (% of those who took each galactagogue)).

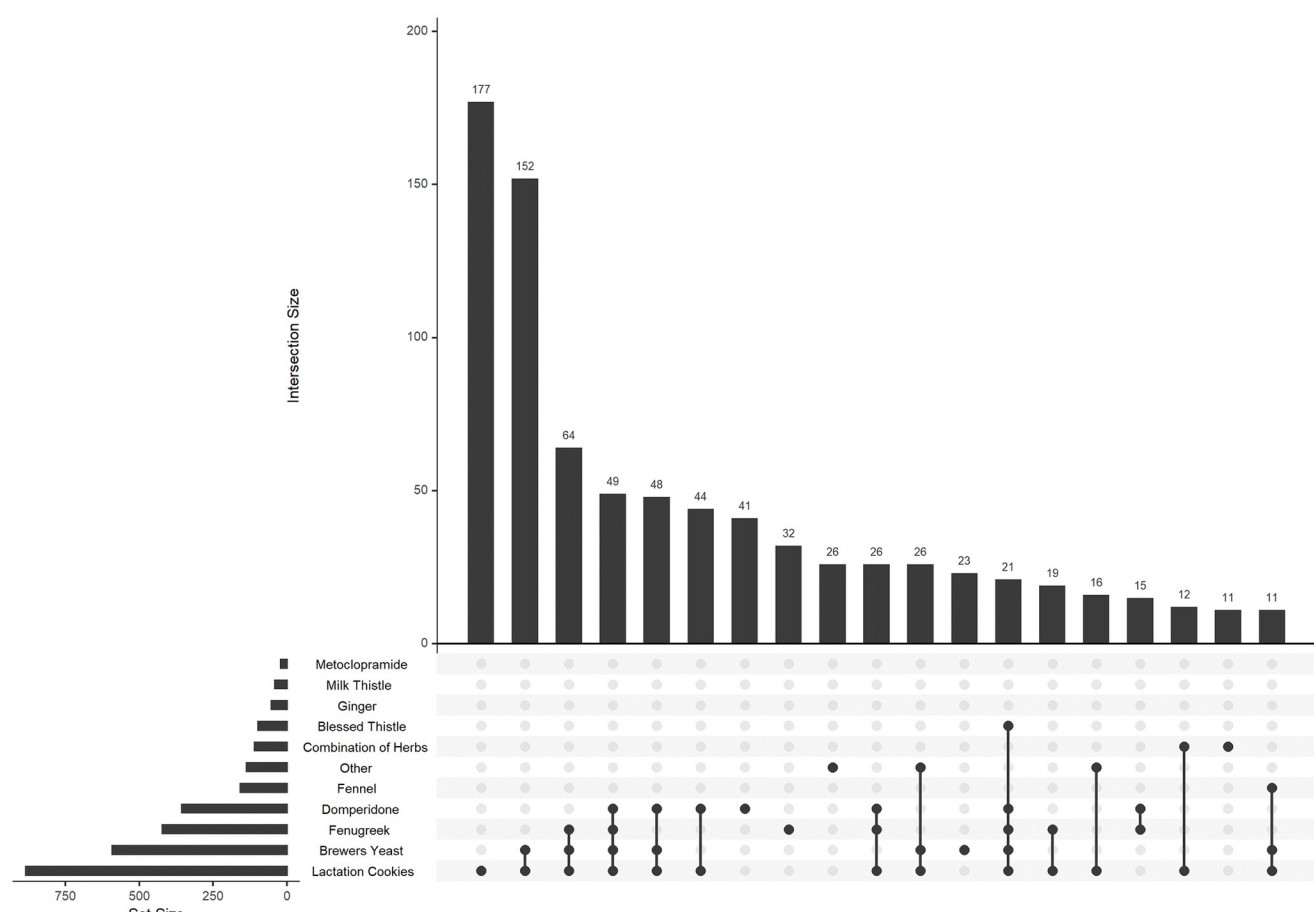

**Fig 1. UpSet plot showing the use of different galactagogues and combinations thereof in breastfeeding women (n = 1120).**

commenced within the first seven days. Timing of commencement varied considerably according to the individual type of galactagogue used. The proportion of women reporting commencing individual galactagogues within the first seven days postpartum, ranged from 4 to 67%.

## Effectiveness

The perceived effectiveness of galactagogues is reported in Fig 3. The mean perceived effectiveness for eight of nine galactagogues was rated as being between 'slightly' (2) and 'moderately' (3) effective (Fig 3), except for domperidone which users reported as having the highest perceived effectiveness (3.3 ± 1.2; mean ± standard deviation).

## Side effects

Side effects women experienced according to galactagogue use are presented in Table 3. Domperidone had the highest proportion of women reporting one or more side effects (45%), compared to less than 20% of women using herbal galactagogues. For domperidone and metoclopramide, 9% and 19% of women respectively stopped taking the medication due to side effects. Greater than 20% of domperidone users experienced two or more side effects.

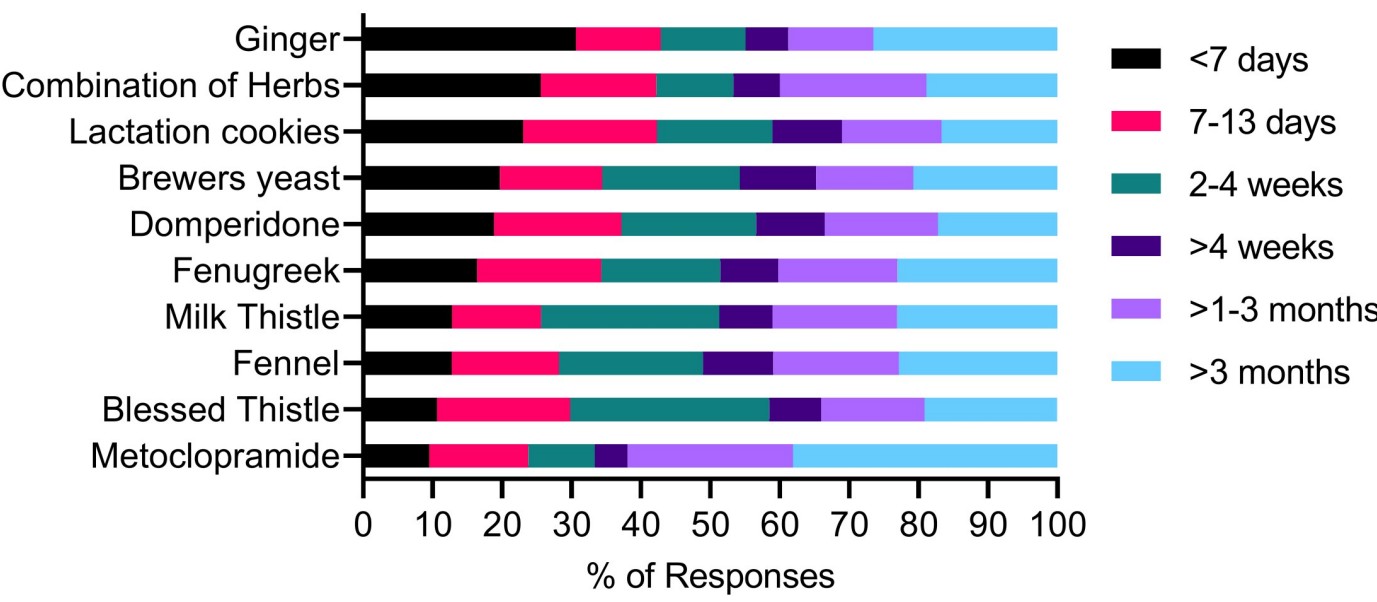

**Fig 2. Timing of commencement postpartum of galactagogues during breastfeeding.**

### Duration of use

The median reported duration of use for each galactagogue is presented in Fig 4. Overall, the median reported duration of use was longer in women who were currently taking a galactagogue at the time of the survey completion. Median durations of use varied from 2 (ginger) to 7 (combination of herbs) weeks for those who had stopped using a substance, and 6 (milk thistle) to 19 weeks (ginger) for those who were continuing use at the time of the survey.

### Recommendations

The percentages of women who would recommend a particular galactagogue to a friend is presented in Fig 5. Overall, 75% would recommend a galactagogue to a friend. There appeared to be a strong correlation between the perceived effectiveness of a galactagogue and whether or not women would recommend it to a friend. Of the 25% of women who would not recommend to a friend, 71% indicated a perceived lack of effectiveness as a reason.

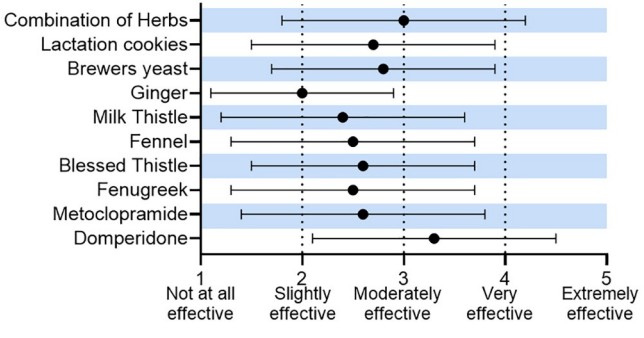

**Fig 3. Perceived effectiveness of galactagogues used by women who were breastfeeding (n = 1120).**

**Table 3. Self-reported side effects for galactagogues used by breastfeeding women (n = 1120).**

| | Domperidone | Metoclopramide | Fenugreek | Blessed thistle | Fennel | Milk thistle | Ginger | Brewer's yeast | Lactation cookies | Combination of herbs |
|---|---|---|---|---|---|---|---|---|---|---|
| Took substance (N) | 355 | 21 | 421 | 98 | 157 | 40 | 52 | 592 | 884 | 109 |
| Any side effects * | 159 (45) | 6 (29) | 72 (17) | 8 (8) | 7 (4) | 4 (10) | 3 (6) | 65 (11) | 110 (12) | 5 (5) |
| Two or more side effects* | 80 (23) | 5 (24) | 22 (5) | 4 (4) | 3 (2) | 3 (8) | 1 (2) | 12 (2) | 15 (2) | 0 (0) |
| Individual side effects* | | | | | | | | | | |
| Weight gain | 88 (25) | 2 (10) | 10 (2) | 1 (1) | | 2 (5) | 1 (2) | 20 (3) | 79 (9) | 1 (1) |
| Headache | 59 (17) | 3 (14) | 9 (2) | 1 (1) | | 1 (3) | 1 (2) | 4 (1) | 4 (0) | 1 (1) |
| Dry mouth | 47 (13) | 3 (14) | 11 (3) | 2 (2) | 1 (1) | 2 (5) | 1 (2) | 9 (2) | 11 (1) | 1 (1) |
| Fatigue | 31 (9) | 3 (14) | 4 (1) | 2 (2) | | | | | 6 (1) | |
| Irritability | 22 (6) | 3 (14) | 2 (0) | | | 1 (3) | | 3 (1) | 4 (0) | 1 (1) |
| Depression | 20 (6) | 2 (10) | 3 (1) | | | | | 3 (1) | 3 (0) | |
| Stomach cramps | 14 (4) | | 17 (4) | 3 (3) | 3 (2) | | | 16 (3) | 13 (1) | |
| Nausea | 13 (4) | 3 (14) | 11 (3) | 3 (3) | 4 (3) | 1 (3) | 1 (2) | 11 (2) | 4 (0) | |
| Heart palpitations /racing heart | 13 (4) | 3 (14) | 3 (1) | | | | | 2 (0) | 2 (0) | |
| Dizziness /fainting | 12 (3) | | 5 (1) | 1 (1) | | 1 (3) | | 4 (1) | 5 (1) | |
| Involuntary movements /jerking | 4 (1) | 2 (10) | | | | | | | | |
| Skin rash | 2 (1) | | 2 (0) | | | | | 1 (0) | 1 (0) | |
| Other | 12 (3) | | 14 (3) | 1 (1) | 2 (1) | | | 7 (1) | 12 (1) | 1 (1) |
| *Body odour* | | | 11 (3) | | | | | | | |
| *Decreased supply* | | | 6 (1) | | | | | | | |
| *Gas/bloating* | | | | | | | | 8 (1) | | |

* n (% of those who took each galactagogue).

## Discussion

In this large contemporary survey of Australian women, galactagogue use was reported by 60% of women at some stage during their lactation. Women commonly reported using multiple galactagogues, with median durations of use from 2–19 weeks or more and 50% of galactagogues being commenced within the first four weeks postpartum. Galactagogues appeared to be well tolerated, except for pharmaceutical galactagogues, where side effects were reported by approximately 50% of women. The widespread utilisation and experiences of galactagogues in postpartum women highlights the importance of future research aimed at (a) understanding why women are using them in the face of limited evidence and guidance about their use, and (b) improving evidence regarding the efficacy and safety of individual galactagogues to support informed decision making.

The need to develop additional strategies to support breastfeeding mothers is reflected in nearly half of our sample reporting that they felt they could not produce enough milk for their child at some stage, with almost 1 in 5 women discontinuing breastfeeding due to concerns about their milk supply. It is uncertain whether concerns related to breast milk supply were real or perceived, determining which has been often recognised as a common challenge within clinical practice settings [6]. The high proportion of women reporting concerns about their breast milk production in our study (approximately 50%) is consistent with previous studies from Australia (45%) and the United States (76%) [12, 30].

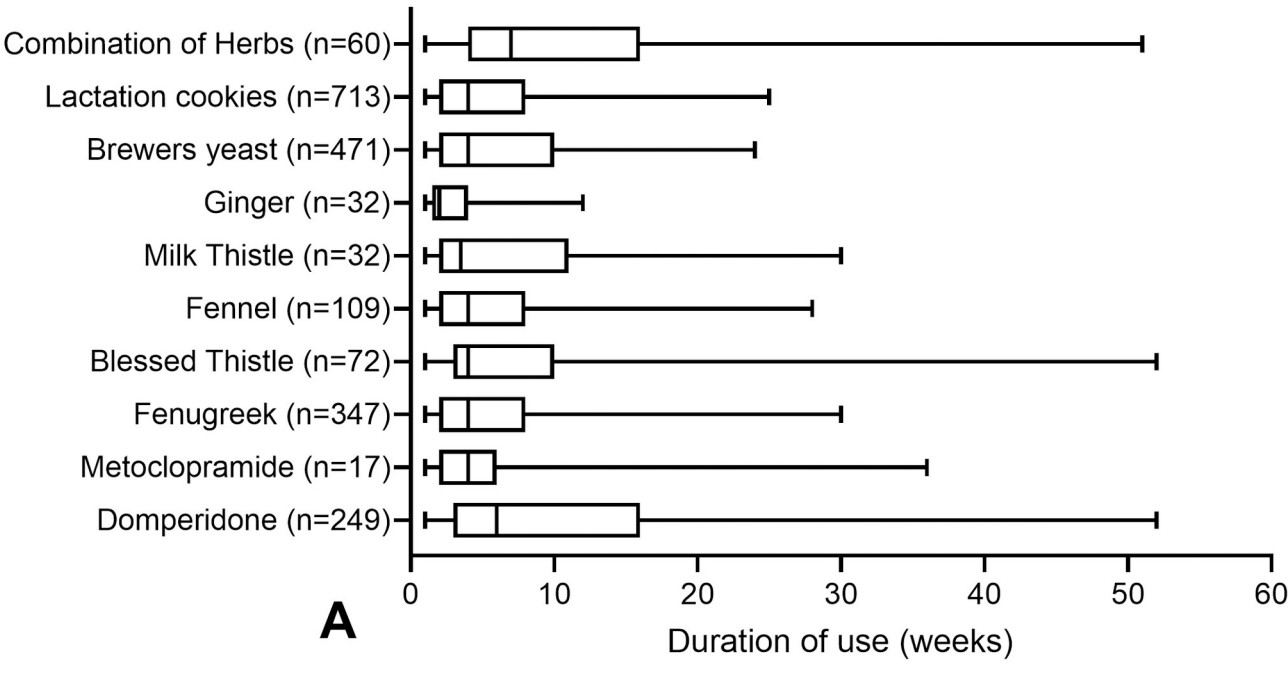

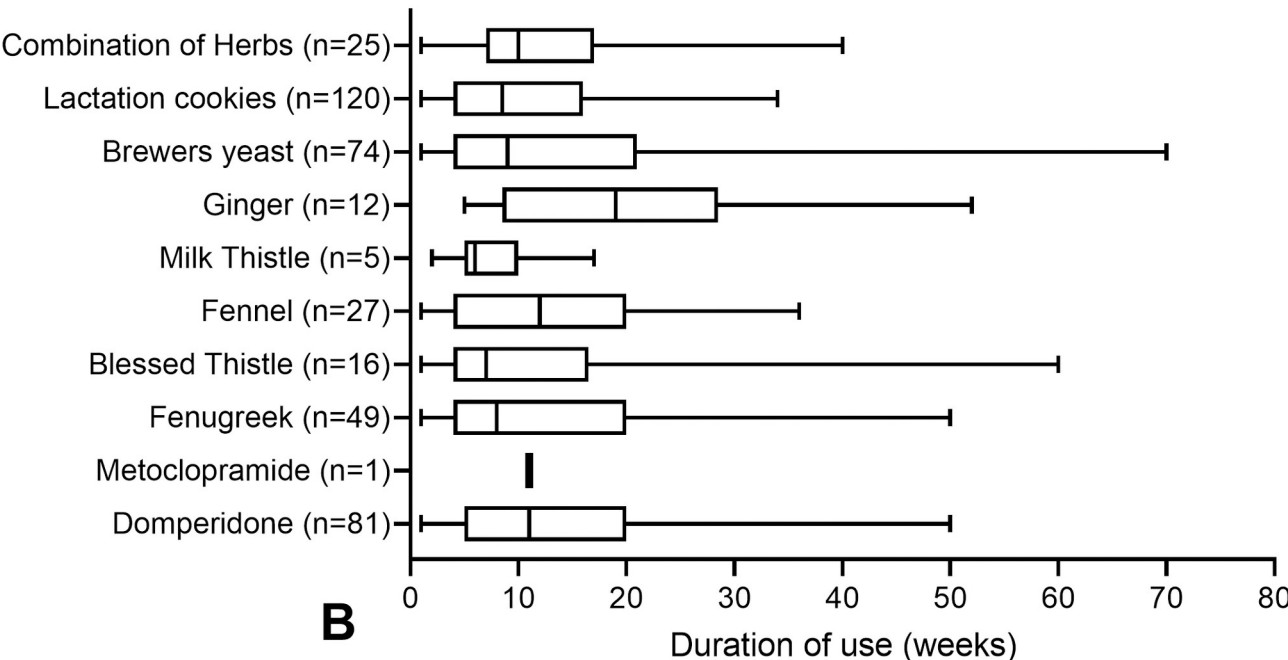

**Fig 4. Duration of galactagogue use by women who had (a) stopped use and those who are (b) continuing use at the time of survey completion (median, inter-quartile range, and 5ᵗʰ to 95ᵗʰ percentile whiskers).**

Our data showed that women were more likely to take a galactagogue based on several pregnancy/birth characteristics such as primiparity, preterm birth or caesarean delivery, as well as perceived low breast milk supply. These risk factors are consistent with those previously reported in the literature, and commonly associated with breastfeeding difficulties [8, 22, 31].

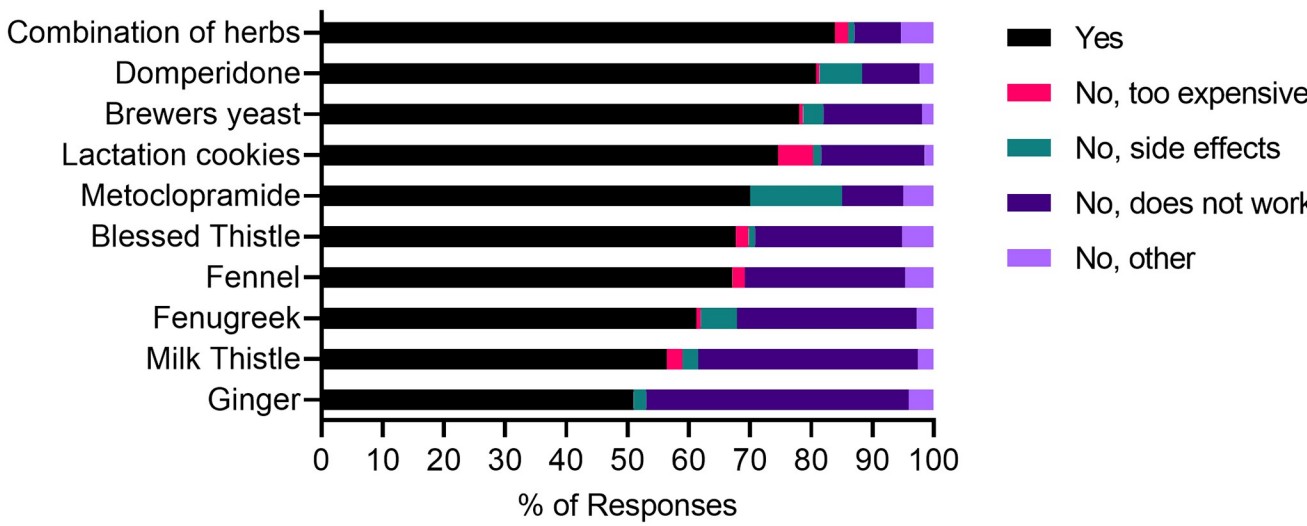

**Fig 5. Women's recommendations of galactagogues to a friend and their reasons for not recommending.**

A previous 2012 survey of women in Western Australia found that 24% of 304 respondents reported using a herbal galactagogue during breastfeeding [18]. The most common galactagogues included fenugreek (18%), blessed thistle (6%) and fennel (5%) [18]. In comparison, a 2015 US survey of 188 women reported herbal galactagogue use in 46% of respondents [12]. The most common galactagogues were fenugreek (46%), fennel (16%) and milk thistle (13%) [12]. However, the survey was restricted to women who reported using or intending to use galactagogues. Notably, neither of these studies collected data regarding the use of the dietary galactagogues lactation cookies or brewer's yeast which were the most commonly reported galactagogues in our survey. Regarding the use of herbal galactagogues, we observed similar high usage of fenugreek, and lower but notable use of fennel [12]. The number of women reporting using domperidone in our survey (1 in 5 respondents) was higher than initially anticipated. A previous Australian audit of domperidone use in the postpartum period at a single tertiary maternity teaching hospital from 2000 to 2010 reported a prevalence of 5% [19]. However, as the audit was restricted to domperidone supplied from the hospital pharmacy department, it likely represents an underestimation of total use [19]. By comparison, international studies evaluating domperidone use from 2011 to 2015 have produced widely varying prevalence ranging from 2% in the UK [20] 2.7% in the US [12], and 20% in Canada [21]. Among high-risk subgroups, such as women with preterm birth, the prevalence of domperidone use increased to 30% [19, 21]. Such differences may reflect differences in inter-country domperidone availability, clinical practice guidelines and prescriber/consumer awareness.

Domperidone had the highest perceived effectiveness rating but also had the highest proportion of women reporting side effects. While previous meta-analyses provide moderate-quality evidence to support the use of domperidone in managing lactation insufficiency following preterm birth [13], there is no such equivalent evidence that it is effective in mothers of otherwise healthy term infants [10]. This represents a significant evidence-gap given widespread uptake of domperidone use following term birth.

While fenugreek was the most commonly used herbal galactagogue and appeared to be well-tolerated, a recent meta-analysis demonstrated that it seems to be no more effective than a placebo in treating lactation insufficiency [10, 32].

The observation that a high proportion of women are taking galactagogues based on recommendations from the internet is consistent with a 2015 survey conducted in the United States demonstrated that 48% of women taking fenugreek sourced their information online. The same study also found that up to 85% of women sought information from sources other than their primary care provider or lactation consultants [12]. Frequent use of information sources other than healthcare professionals raises concerns regarding whether or not women are being provided with evidence-based information regarding the use of galactagogues to support informed decision making. This is backed up by findings from an Australian survey of women's attitudes to herbal medicine during lactation that found that while the internet was again a common source of information, women often doubted the reliability of information from the internet and cited the need for information and resources endorsed by reputable breastfeeding organisations and healthcare professionals [30, 33]. The second highest source of information was women's friends, which may suggest that women prioritise others' anecdotal experiences over that of evidence-based resources or trained health care professionals.

The fact that 1 in 6 respondents started using various galactagogues within the first seven days postpartum raises potential concerns, particularly given the challenge of assessing the adequacy of breast milk production in the early postpartum period [34, 35]. These findings may indicate that women may be turning to galactagogues prophylactically (without actually having low breast milk supply) or using them as early treatments before trying non-pharmacological strategies. While 71% of women reported seeing a lactation consultant, we do not know when this occurred relative to the commencement of galactagogues. The observation that 20% of galactagogue use occurred after three months postpartum highlights the importance of continued breastfeeding support beyond the immediate postpartum period.

## Strengths and limitations

This survey is the first to examine galactagogue use, the timing and duration of use, as well as perceived effectiveness and side effects of common galactagogues in the community. This study has several limitations. Our survey used non-probabilistic sampling and snowballing sampling techniques, making it difficult to extrapolate findings to the broader Australian population of breastfeeding women. Based on national Australian perinatal statistics, while our survey included a higher proportion of women born in Australia (86% vs 65%), the distribution of other key demographic and birth characteristics known to influence breastfeeding outcomes such as maternal age at delivery (31.5 years vs. 30.4 years), delivery by caesarean section (33% vs. 35%), preterm birth (12% vs. 9%), and prevalence of overweight/obese (49% vs 45%), were similar [36].

The survey measured women's perceived effectiveness and did not utilise objective measures of changes in breast milk supply. The support and clinical care women received during breastfeeding were also not reported, meaning some women's perceived increase in supply may have been unrelated to galactagogue use.

Lactation cookies were provided as one of ten listed galactagogues that women could choose from. However participants were not provided with a definition of lactation cookies nor a pre-defined list of ingredients. As such, our survey did not account for possible variations in ingredients between different lactation cookies, and there is likely to be some crossover with other galactagogues listed in the survey, particularly brewer's yeast which is one of the most common ingredients in lactation cookies. While 1217 women were breastfeeding at the time of the survey, responses were obtained from 657 women who had completed breastfeeding up to an average of 3.9 years before completing the survey. This raises the possibility of errors related to women's ability to recall exact timings and durations of use correctly.

However, previous studies suggest the degree of error is likely to be small, with a 1-month error in reporting among women recalling information from 1 to 3.5 years prior [37]. Furthermore, for women continuing to breastfeed and taking a galactagogue at the time of completing the survey, it is not possible to correctly define the total duration of use or their complete set of experiences. Lastly, we did not ask about galactagogue use before birth. Some women use herbal galactagogues before birth to stimulate lactation initiation, which is of concern as some popular herbal galactagogues such as fenugreek and milk thistle may cause uterine contractions [38].

## Concluding remarks

This large online survey demonstrates that the use of galactagogues appears to be very common in Australia. Women seem to be using multiple galactagogues during breastfeeding, with evidence of frequent initiation in the first week postpartum and long durations of use. The incidence of side effects appeared higher for women taking pharmaceutical agents compared to herbal galactagogues. However, a number of side effects were still reported by women using herbal or food-based galactagogues, suggesting they are not completely benign. The high prevalence of women taking galactagogues based on recommendations obtained through the internet or friends, rather than healthcare providers, raises concerns surrounding the potential quality of the information they receive, particularly in light of the lack of evidence surrounding the effectiveness and safety of most galactagogues. Overall, our findings highlight the need for further high-quality research, particularly appropriately powered randomized controlled trials, to generate robust evidence about the efficacy and safety of galactagogues to support evidence-based strategies to improve breastfeeding outcomes.

## Supporting information

**S1 File. Boosting breast milk supply survey.** All survey questions asked regarding women's use and experiences with substances to boost breast milk supply.
(PDF)

## Author Contributions

**Conceptualization:** Gabriella Zizzo, Alice R. Rumbold, Lisa H. Amir, Luke E. Grzeskowiak.

**Formal analysis:** Grace M. McBride, Luke E. Grzeskowiak.

**Funding acquisition:** Gabriella Zizzo, Alice R. Rumbold, Lisa H. Amir, Luke E. Grzeskowiak.

**Investigation:** Grace M. McBride, Robyn Stevenson, Gabriella Zizzo, Alice R. Rumbold, Lisa H. Amir, Luke E. Grzeskowiak.

**Visualization:** Grace M. McBride, Luke E. Grzeskowiak.

**Writing – original draft:** Grace M. McBride.

**Writing – review & editing:** Grace M. McBride, Robyn Stevenson, Gabriella Zizzo, Alice R. Rumbold, Lisa H. Amir, Amy K. Keir, Luke E. Grzeskowiak.

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
